



# Global trends in marine nitrate N isotopes from observations and a neural network-based climatology

**Patrick A. Rafter**[1], **Aaron Bagnell**[2], **Dario Marconi**[3], **and Timothy DeVries**[2]

[1]Department of Earth System Science, University of California, Irvine, CA, USA
[2]Department of Geography, University of California, Santa Barbara, CA, USA
[3]Department of Geosciences, Princeton University, Princeton, NJ, USA

**Correspondence:** Patrick A. Rafter (prafter@uci.edu)

**Abstract.** Nitrate is a critical ingredient for life in the ocean because, as the most abundant form of fixed nitrogen in the ocean, it is an essential nutrient for primary production. The availability of marine nitrate is principally determined by biological processes, each having a distinct influence on the N isotopic composition of nitrate (nitrate $\delta^{15}$N) – a property that informs much of our understanding of the marine N cycle as well as marine ecology, fisheries, and past ocean conditions. However, the sparse spatial distribution of nitrate $\delta^{15}$N observations makes it difficult to apply this useful property in global studies or to facilitate robust model–data comparisons. Here, we use a compilation of published nitrate $\delta^{15}$N measurements ($n = 12\,277$) and climatological maps of physical and biogeochemical tracers to create a surface-to-seafloor, 1° resolution map of nitrate $\delta^{15}$N using an ensemble of artificial neural networks (EANN). The strong correlation ($R^2 > 0.87$) and small mean difference ($< 0.05\,\%_o$) between EANN-estimated and observed nitrate $\delta^{15}$N indicate that the EANN provides a good estimate of climatological nitrate $\delta^{15}$N without a significant bias. The magnitude of observation-model residuals is consistent with the magnitude of seasonal decadal changes in observed nitrate $\delta^{15}$N that are not captured by our climatological model. As such, these observation-constrained results provide a globally resolved map of mean nitrate $\delta^{15}$N for observational and modeling studies of marine biogeochemistry, paleoceanography, and marine ecology.

# 1 Introduction

In contrast to other marine nutrients (e.g., phosphate and silicate), the inventory of nitrate ($NO_3^-$) and other fixed nitrogen (N) is mediated by biological processes, where the main source is $N_2$ fixation by diazotrophic phytoplankton and the main sink is denitrification (via a microbial consortium in oxygen deficient waters and sediments) (Codispoti and Christensen, 1985). Biological processes also determine the distribution of marine nitrate throughout the water column, with phytoplankton assimilating nitrate and the lowering of nitrate concentrations in the surface ocean and the microbially mediated degradation of organic matter in the subsurface. The latter involves the multi-step process of ammonification (organic matter $\rightarrow NH_4^+$) and nitrification ($NH_4^+ \rightarrow NO_2^- \rightarrow NO_3^-$). By regulating the global inventory and distribution of marine nitrate, these N cycling processes control global net primary productivity, the transfer of nutrients to higher trophic levels such as fishes, and the strength of the ocean's biological carbon pump (Dugdale and Goering, 1967).

Each of these biologically mediated N transformations affects the N isotopic composition of nitrate in unique ways (Fig. 1a and b, and see Sect. 2), adjusting the relative abundance of $^{15}$N and $^{14}$N in oceanic nitrate relative to the atmosphere. ($\delta^{15}$N = ($^{15}$N/$^{14}$N$_{sample}$/$^{15}$N/$^{14}$N$_{standard}$) − 1, multiplied by 1000 to give units of per mill (‰); see Sigman and Casciotti, 2001, for simplified equations from Mariotti et al., 1981.) Nitrate $\delta^{15}$N measurements have become a powerful tool for understanding the "biogeochemical history" of marine nitrate (Rafter et al., 2012), which includes nitrate assimilation by phytoplankton (Miyake and Wada, 1967; Wada and

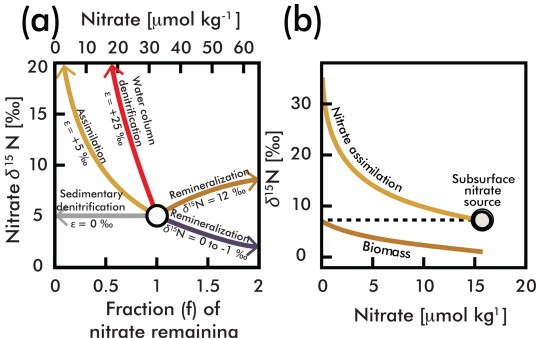

**Figure 1.** **(a)** A comparison of influences on average deep-sea nitrate (circle; concentration and $\delta^{15}$N estimated here by EANN results in Table 1) including the isotope effects of assimilation (yellow arrow), water column and sedimentary denitrification (red and gray arrows), and the addition of nitrate via remineralization of organic matter with higher and lower $\delta^{15}$N (brown and purple arrows) (modified from Galbraith et al., 2008). **(b)** An example of N isotopic fractionation on nitrate and organic matter biomass during nitrate assimilation in eastern equatorial Pacific surface waters (from Rafter and Sigman, 2016). These calculations are based on isotopic fractionation equations of Mariotti et al., (1981) simplified in Sigman and Casciotti (2001) with an isotope effect ($\varepsilon$) as shown in **(a)**. The variable $f$ is the initial nitrate concentration.

Hattori, 1978), nitrogen fixation (Carpenter et al., 1997; Hoering and Ford, 1960), denitrification (Liu, 1979), and nitrification (Casciotti et al., 2013). For example, the consumption of nitrate by denitrification (red line in Fig. 1a) has a larger
impact on the residual nitrate $\delta^{15}$N than partial nitrate assimilation by phytoplankton(yellow line in Fig. 1), and thus very high $\delta^{15}$N values serve as a fingerprint of denitrification. Nitrate $\delta^{15}$N is also influenced by the addition of nitrate via remineralization of organic matter. The exact influence
of remineralization depends on the isotopic composition of the organic matter, and could result in higher or lower nitrate $\delta^{15}$N (Fig. 1a). Nitrate introduced into the water column by the remineralization of organic matter formed by $N_2$-fixing phytoplankton has an isotopic composition close to that of
air (0 ‰–1 ‰), and serves to lower the mean ocean nitrate $\delta^{15}$N (Fig. 1b). On the other hand, organic matter formed from nitrate assimilation in regions where the plankton use most of the available nitrate can be isotopically heavy, and its remineralization will increase the $\delta^{15}$N of ambient nitrate
(Fig. 1b). The actual value of organic matter $\delta^{15}$N formed from nitrate assimilation is mostly determined by the following: (1) the $\delta^{15}$N of nitrate delivered to the euphotic zone (the subsurface source), which in turn is dependent on the degree of water column denitrification and CEI (2) the de-
gree of nitrate consumption at the ocean surface, with heavier values associated with greater nitrate consumption (Fig. 1b). Accordingly, changes in organic matter $\delta^{15}$N (and therefore sediment $\delta^{15}$N used for paleoceanographic work) can reflect variability of the source nitrate $\delta^{15}$N and/or variability of the

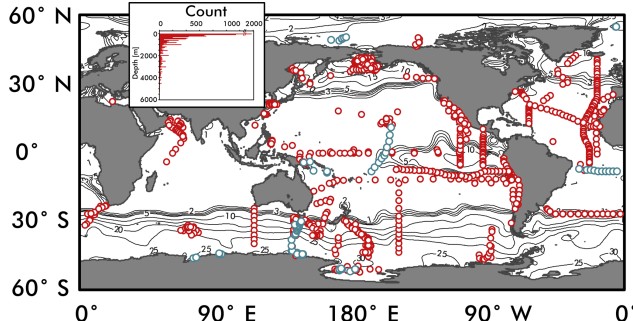

**Figure 2.** The location of global nitrate $\delta^{15}$N observations used to constrain the ensemble of artificial neural networks are shown as red circles. Observations used as an "external validation dataset" (those withheld from training the EANN) are shown in blue. Inset figure shows the number of observations versus depth. Contours are surface nitrate concentrations for October–December from World Ocean Atlas (Garcia et al., 2010).

degree of nitrate consumption (e.g., see Rafter and Charles, 30 2012).

Because of nitrate's place at the base of the marine ecosystem, nitrate $\delta^{15}$N is also useful for understanding the life-cycles of higher trophic level organisms such as fish (Graham et al., 2007; Tawa et al., 2017) and fishery productivity 35 (Finney et al., 2002, 2000). The $\delta^{15}$N of whole sediment and microfossils provides insight by proxy of past ocean nitrate transformations (Altabet and Francois, 1994a; Kienast et al., 2008; Ren et al., 2009; Robinson et al., 2004; Sigman et al., 1999b) – work that places important constraints on modern 40 ocean N cycling (Altabet, 2007; Eugster et al., 2013; Ren et al., 2017). With an understanding of the N transformations described above and their influences on the N isotopic composition of nitrate, we can begin using nitrate $\delta^{15}$N measurements to trace the integrated biogeochemical history of 45 marine nitrate. However, identifying basin- and global-scale trends in nitrate $\delta^{15}$N is challenged by the limited spatial extent of nitrate $\delta^{15}$N observations (Fig. 2). Here, we compile a global database of nitrate $\delta^{15}$N measurements (Fig. 2) and use an ensemble of artificial neural networks (EANN) to pro- 50 duce a map of the global nitrate $\delta^{15}$N distribution at 1° spatial resolution. We find that the mapped nitrate $\delta^{15}$N climatology matches the observations well and should be a valuable tool for estimating mean conditions and for constraining predictive nitrate $\delta^{15}$N models (Somes et al., 2010; Yang and 55 Gruber, 2016). Below we briefly discuss how the EANN was used to produce global maps of nitrate $\delta^{15}$N (Sect. 2), address the ability of the EANN to match the measured $\delta^{15}$N (Sect. 3), and examine the EANN-mapped $\delta^{15}$N climatology and global compilation of nitrate $\delta^{15}$N in the context of pub- 60 lished work (Sect. 4).

## 2 Methods

### 2.1 Data compilation

Nitrate $\delta^{15}$N observations (Fig. 2; references and region of measurement in Appendix) were compiled from studies dating from 1975 (Cline and Kaplan, 1975) to 2018 (Fripiat et al., 2018), including data from the GEOTRACES intermediate data product (Schlitzer et al., 2018). Whenever possible, the data were acquired via the original author, but in other cases the data were estimated from the publication directly. All observations were treated equally, although the failure to remove nitrite when using the "denitrifier method" may bias the nitrate $\delta^{15}$N to low values (Rafter et al., 2013). These measurements have been identified as "nitrate + nitrite" in the dataset to acknowledge this potential biasing, which predominantly affects observations in the upper 100 m (Kemeny et al., 2016; Rafter et al., 2013).

### 2.2 Building the neural network model

We utilize an ensemble of artificial neural networks (EANN) to interpolate our global ocean nitrate $\delta^{15}$N database (Fig. 2), producing complete 3-D maps of the data. By utilizing an artificial neural network (ANN), a machine learning approach that effectively identifies nonlinear relationships between a target variable (the isotopic dataset) and a set of input features (other available ocean datasets), we can fill holes in our data sampling coverage of nitrate $\delta^{15}$N.

#### 2.2.1 Binning target variables (Step 1)

We binned the nitrate $\delta^{15}$N observations (red symbols in Fig. 2) to the World Ocean Atlas 2009 (WOA09) grid with a 1° spatial resolution and 33 vertical depth layers (0–5500 m) (Garcia et al., 2010). When binning vertically, we use the depth layer whose value is closest to the observation's sampling depth (the first depth layer has a value of 0 m, the second of 10 m, and the third of 20 m, so all nitrate isotopic data sampled between 0 and 5 m fall in the 0 m bin; between 5 and 15 m they fall in the 10 m bin, etc.). An observation with a sampling depth that lies right at the midpoint between depth layers is binned to the shallower layer. If more than one raw data point falls in a grid cell we take the average of all those points as the value for that grid cell. Certain whole ship tracks of nitrate $\delta^{15}$N data were withheld from binning to be used as an independent validation set (see Sect. 2.2.4).

#### 2.2.2 Obtaining input features (Step 2)

Our input dataset contains a set of climatological values for physical and biogeochemical ocean parameters that form a non-linear relationship with the target data. We have six input features including objectively analyzed annual-mean fields for temperature, salinity, nitrate, oxygen, and phosphate taken from the WOA09 (https://www.nodc.noaa.gov/OC5/WOA09/woa09data.html, last access: 1 January 2017) at 1° resolution. Additionally, daily chlorophyll data from Modis Aqua for the period 1 January 2003 through 31 December 2012 are averaged and binned to the WOA09 grid (as described in Step 1) to produce an annual climatological field of chlorophyll values, which we then log transform to reduce their dynamic range.

The choice of these specific input features was dictated by our desire to achieve the best possible $R^2$ value on our internal validation sets (Step 4). Additional inputs besides those we included, such as latitude, longitude, silicate, euphotic depth, or sampling depth either did not improve the $R^2$ value on the validation dataset or degraded it, indicating that they are not essential parameters for characterizing this system globally. By opting to use the set of input features that yielded the best results for the global oceans, we potentially overlooked combinations of inputs that perform better at regional scales. However, given the scarcity of $\delta^{15}$N data in some regions, it is not possible to ascribe the impact of a specific combination of input features versus the impact of available $\delta^{15}$N data, which may not be representative of the region's climatological state, to the relative model performance in these regions.

#### 2.2.3 Training the ANN (Step 3)

The architecture of our ANN consists of a single hidden layer, containing 25 nodes, that connects the biological and physical input features (discussed in Step 2) to the target nitrate isotopic variable (as discussed in Step 1). The role of the hidden layer is to transform input features into new features contained in the nodes. These are given to the output layer to estimate the target variable, introducing nonlinearities via an activation function. The number of nodes in this hidden layer, as well as the number of input features, determines the number of adjustable weights (the free parameters) in the network. Because there is a danger of over-fitting the model, which occurs when the ANN is over-trained on a dataset so that it cannot generalize well when presented with new data, it is a good practice to have a large number of training data (we have 7170 binned data points) relative to the number of weights (we have 201 free parameters) (Weigend et al., 1990). To create a nonlinear system, an activation function transforms the product of the weights and input features and creates the values assigned to nodes in the hidden layer. These act as new features for estimating the target $\delta^{15}$N data. Our model utilizes the hyperbolic tangent as its activation function between the input and hidden layer as well as between the hidden and output layer due to its speed and general performance (Thimm and Fiesler, 1997).

The values of nodes in the hidden layer ($\mathbf{H}$) can be defined as

$$\mathbf{H} = a(\mathbf{I} \cdot \mathbf{W}_1 + \boldsymbol{b}_1), \tag{1}$$

where $\mathbf{H}$ is an array containing the values of the hidden nodes, $a$ is the activation function. Here the hyperbolic tangent, $\mathbf{I}$, is a $7170 \times 6$ array containing the values of the input features at the locations of the binned observations (there are 7170 binned observations and six input parameters). $\mathbf{W}_1$ is a $6 \times 25$ array of weights that connect input features to hidden nodes, and $\boldsymbol{b}_1$ is a $7170 \times 25$ array of weights (25 unique values repeated 7170 times) that connects a bias node to the hidden nodes. The factor of 25 represents the number of nodes in the hidden layer, chosen by experimentation to find the maximum number of effective parameters (Foresee and Hagan, 1997), i.e., where adding new parameters no longer improves performance on an internal validation set (Step 4). The bias node acts as an offset term, similar to a constant term in a linear function, and has a value that is always one.

At the output layer, the network produces a prediction of the target nitrate isotopic data ($\delta^{15}\mathrm{N}_{\mathrm{pred}}$). Similar to how nodes in the hidden layer are a function of the inputs and a set of weights, $\delta^{15}\mathrm{N}_{\mathrm{pred}}$ is a function of the hidden nodes and an additional set of weights. The predicted values can be defined as

$$\delta^{15}\mathrm{N}_{\mathrm{pred}} = a(\mathbf{H} \cdot \mathbf{W}_2 + \boldsymbol{b}_2), \tag{2}$$

where $\mathbf{H}$ (size $7170 \times 25$) has been previously defined, $\mathbf{W}_2$ (size $25 \times 1$) is a matrix of weights that connect features in the hidden layer to nodes in the output layer, and $\boldsymbol{b}_2$ (size $7170 \times 1$) is an array of weights (all of the same value) that connects a bias node to the output layer.

The ANN learns by comparing $\delta^{15}\mathrm{N}_{\mathrm{pred}}$ to the actual $\delta^{15}\mathrm{N}$ data ($\delta^{15}\mathrm{N}_{\mathrm{data}}$), attempting to minimize the value of the cost function

$$\mathrm{cost} = \frac{\sum\limits_{i=1}^{n} \left(\delta^{15}\mathrm{N}_{\mathrm{pred}}{}^{i} - \delta^{15}\mathrm{N}_{\mathrm{data}}{}^{i}\right)^2}{n} \tag{3}$$

by iteratively adjusting the weights using the Levenberg–Marquardt algorithm (Marquardt, 1963) as a way of propagating the errors between $\delta^{15}\mathrm{N}_{\mathrm{pred}}$ and $\delta^{15}\mathrm{N}_{\mathrm{data}}$ backwards though the network (Rumelhart et al., 1986).

### 2.2.4 Validating the ANN (Step 4)

To ensure good generalization of the trained ANN, we randomly withhold 10 % of the $\delta^{15}\mathrm{N}$ data to be used as an internal validation set for each network. These are data that the network never sees, meaning it does not factor into the cost function, so it works as a test of the ANN's ability to generalize. This internal validation set acts as a gatekeeper to prevent poor models from being accepted into the ensemble of trained networks (see Step 5). A second, independent or "external" validation set (blue symbols in Fig. 2), composed of complete ship transects from the high- and low-latitude ocean were omitted from binning in Step 1 and used to establish the performance of the entire ensemble (Sect. 3.2). Our

rationale for using complete ship transects is the following. If we randomly choose 10 % of observations to perform an external validation, this dataset will be from the same cruises as the wider data. In other words, despite being randomly selected, the validating observational dataset will be highly correlated geographically. Contrast this with validating the EANN results with observations from whole research cruises in unique geographic regions – areas where the model has not "learned" anything about nitrate. We therefore argue that observations from whole ship tracks provide a more rigorous test of the model.

### 2.2.5 Forming the ensemble (Step 5)

The ensemble is formed by repeating Steps 3 to 4 (using a different random 10 % validation set) until we obtain 25 trained networks for the nitrate $\delta^{15}\mathrm{N}$ dataset. A network is admitted into the ensemble if it yields an $R^2$ value greater than 0.81 on the validation dataset. Using an EANN instead of any single network provides several advantages. For example, the random initialization of the weight values in each network as well as differences in the training and internal validation sets used across members make it possible for many different networks to achieve similar performance on their respective validation set while generalizing to areas with no data coverage differently. By performing this type of data subsampling and taking an ensemble average, similar to bootstrap aggregating (Breiman, 1996), this approach on average improves the robustness of the generalization in areas without data coverage compared to a single randomly generated ensemble member. Compared to each of its members, our ensemble mean sees improved performance on all internal validation sets and has a higher $R^2$ and lower root mean square error on the independent validation set compared to 19 of the 25 members. The range of values given by the ensemble also provides a measure of the uncertainty for our estimations of $\delta^{15}\mathrm{N}$.

## 3 Results

### 3.1 Global nitrate $\delta^{15}\mathrm{N}$ observations

The global compilation of nitrate $\delta^{15}\mathrm{N}$ includes 1180 stations from all major ocean basins and some minor seas (Fig. 2) giving a total of 12 277 nitrate $\delta^{15}\mathrm{N}$ measurements. Within this dataset, 1197 nitrate $\delta^{15}\mathrm{N}$ measurements were withheld from the EANN and used to validate the EANN results (the "external" validation dataset; blue symbols in Fig. 2, see Sect. 2). With observations from the surface to as deep as 6002 m (Rafter et al., 2012), we find that nitrate $\delta^{15}\mathrm{N}$ ranges from $\approx 1\,\%_0$ in the North Atlantic (e.g., Marconi et al., 2015) to 68.7 $\%_0$ in the eastern tropical South Pacific (Bourbonnais et al., 2015). Nitrate $\delta^{15}\mathrm{N}$ of $\approx 1\,\%_0$ was also irregularly observed in the shallow North and South Pacific (Liu et al., 1996; Yoshikawa et al., 2015). These latter observations were

included in the training dataset, although we should note that measurements using the "Devarda's alloy" method (Liu et al., 1996) are thought to be biased towards lower values (Altabet and Francois, 2001). Similarly, the inclusion of nitrite during "denitrifier method" nitrate $\delta^{15}$N measurements can bias the measurement towards lower values (Kemeny et al., 2016; Rafter et al., 2013).

### 3.2 Marine nitrate $\delta^{15}$N observations model comparison

The observed and EANN-predicted nitrate $\delta^{15}$N measurements are distributed around a 1 : 1 line in Fig. 3a (all data), with considerably less scatter for the values > 1000 m depth (Fig. 3b). The correlation coefficient of determination for the observations versus the model nitrate $\delta^{15}$N gives an $R^2 =$ 0.75 for the raw and unbinned observations used to train the EANN (see Fig. 3a) and an $R^2$ of 0.78 for the external validation dataset. The model matches the observations even better for depths > 1000 m (Fig. 3b) with an $R^2 = 0.83$ (values used in model) and $R^2 = 0.78$ for the external validation dataset. We can also examine the performance of the EANN with the nitrate $\delta^{15}$N "residual" or the difference between observed and modeled $\delta^{15}$N (Fig. 3c), which indicates a mean residual or "mean bias" value of $-0.03$‰ and $+0.18$‰ for the model training and external validation datasets, respectively.

Examining the observation-EANN residuals via the root mean square error (RMSE; Fig. 3d), we find an RMSE of 1.94‰ for the data used to train the EANN and an RMSE of 1.26‰ for the external validation dataset. There is a clear relationship between the residual, the RMSE and depth, with significantly higher values for the upper 500 m (Fig. 3c and d) and a RMSE of about 0 below 500 m. Comparing these residual values with dissolved oxygen concentrations (color in Fig. 3c), we find that a > 2‰ RMSE for the surface (Fig. 3d) is associated with high oxygen while > 2.7‰ RMSE at ≈ 250 m is associated with the lowest oxygen (color in Fig. 3c).

The RMSE patterns in Fig. 3c and d are to be expected given the natural variability in nitrate $\delta^{15}$N driven by assimilation in the upper ocean and denitrification in the shallow subsurface – variability which is not captured by the climatological EANN. The CE2 only available nitrate $\delta^{15}$N timeseries is by Rafter and Sigman (2016) in the eastern equatorial Pacific, and it shows that variability of nitrate assimilation produces seasonal-to-interannual deviations of $\delta^{15}$N of ±2.5‰, which is similar to the magnitude of the RMSE in the surface ocean (2.2‰). Although there are no nitrate $\delta^{15}$N time series measurements from the subsurface oxygen-deficient zone (ODZ) waters where denitrification occurs, nitrate $\delta^{15}$N in ODZs presumably have similar seasonal-to-interannual (or longer timescale) variability due to changes in the rate and extent of water column denitrification (Deutsch et al., 2011; Yang et al., 2017). For example, a larger degree

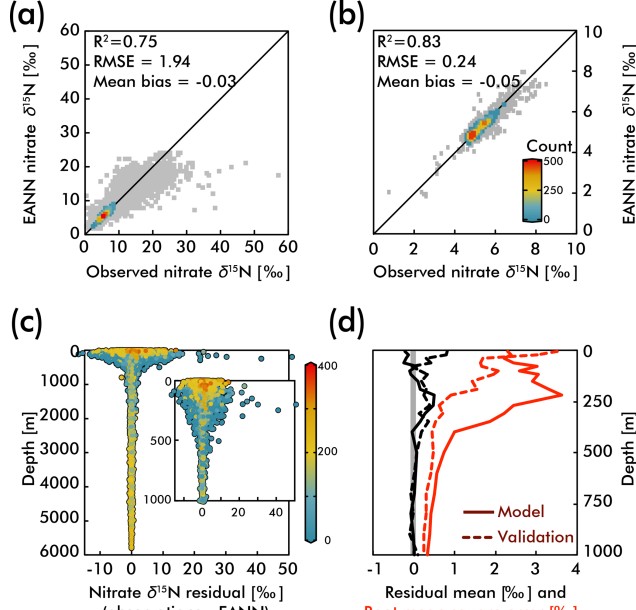

**Figure 3.** The observed versus EANN-predicted nitrate $\delta^{15}$N are shown for all data at all depths in **(a)** and for > 1000 m in **(b)**. The colors indicate the number of comparisons on the World Ocean Atlas grid (legend in **b**; gray = 1 observation). The anomalously high observed nitrate $\delta^{15}$N values (> 30‰) in **(a)** are exclusively from the eastern tropical South Pacific waters (Bourbonnais et al., 2015; Casciotti et al., 2013; Rafter et al., 2012; Ryabenko et al., 2012). The difference (or residual) between the observations and EANN nitrate $\delta^{15}$N is shown for all depths and the upper 1000 m in **(c)** with colors representing the dissolved oxygen content. Note the largest offsets between 100 and 500 m in **(c)** are associated with lowest oxygen content. Similarly, the mean residual (black) and root mean square error (RMSE; red) with depth **(d)** are highest in the upper 500 m. Dashed lines in **(d)** demonstrate the same statistics, but for the external validation dataset (blue in Fig. 2).

of nitrate undergoing water column denitrification would explain the extreme $\delta^{15}$N values at the bottom right of Fig. 3a – observations that all come from the ODZ waters of the eastern tropical South Pacific (Bourbonnais et al., 2015; Casciotti et al., 2013; Rafter et al., 2012; Ryabenko et al., 2012). Some of these very high nitrate $\delta^{15}$N values are associated with nitrate concentrations < 1 µmol kg$^{-1}$ (Bourbonnais et al., 2015), values much lower than those within our climatology for the subsurface eastern tropical South Pacific. These values thus represent episodic denitrification events that the EANN will not be able to capture because it is trained on climatological data. In the deep ocean where temporal variability is smaller, the observation-EANN residuals of 0.2‰ are the same magnitude as the $\delta^{15}$N analytical errors, further emphasizing the ability of the model to match climatological average conditions.

## 4   Discussion

The EANN's skillful estimate of climatological nitrate $\delta^{15}$N will be useful for studies of the marine nitrogen cycle. The zonal average view of EANN nitrate $\delta^{15}$N for each major
ocean basin (Fig. 4) includes statistics comparing the observations to the EANN results above and below 1000 m. These region-specific statistics show a weaker correlation between EANN and observed nitrate $\delta^{15}$N in the deep Atlantic and Southern Ocean, despite a low RMSE and negligible mean
bias. This weak correlation likely derives from the limited variability of deep nitrate $\delta^{15}$N ($\pm 0.1$‰) in these basins (see Fig. 5d).

The nitrate $\delta^{15}$N sections in Fig. 4 show elevated values for the low latitude, upper mesopelagic Pacific Ocean (Fig. 4a),
and Indian Ocean (Fig. 4d) where water column denitrification raises the residual nitrate $\delta^{15}$N (Fig. 1a). Viewing this elevated nitrate $\delta^{15}$N at the 250 m depth horizon (Fig. 5) better reveals the spatial heterogeneity of the observations and EANN results. (It is because of this intra-basin heterogene-
ity, and the fact that many observations are biased towards the areas of denitrification, that we did not plot the observed nitrate $\delta^{15}$N within the zonally averaged Fig. 4 views.) The EANN error for the Fig. 5 depth intervals (Fig. 5e–h) is the standard deviation of the 25 ensemble members of the EANN
and shows a decrease in ensemble variability with depth – a trend that is consistent with the overall decrease in observed nitrate $\delta^{15}$N variability with depth (Figs. 4 and 5).

Below we inspect the observed and EANN-predicted nitrate $\delta^{15}$N and discuss the consistency of these results with
30 our understanding of published work. This analysis begins with the spatial distribution of nitrate delivered to the upper ocean. We then discuss the impacts of upper ocean nitrate assimilation on organic matter $\delta^{15}$N and consider the influence of organic matter remineralization on subsurface nitrate.

### 4.1   Subsurface and surface nitrate $\delta^{15}$N

The nitrate $\delta^{15}$N distribution at 250 m depth (Fig. 5b) offers a view of nitrate at a depth that is deeper than source waters in many ocean regions (e.g., 100 to 150 m in the equatorial Pacific; Rafter and Sigman, 2016), but is negligibly influenced
by nitrate assimilation, and therefore provides a qualitative view of spatial trends in nitrate delivered to the surface ocean. Nitrate $\delta^{15}$N at this depth is highest in the northeastern and southeastern tropical Pacific and the Arabian Sea (Fig. 5b), due to the influence of water column denitrification in the
ODZs in these regions (Altabet et al., 2012; Bourbonnais et al., 2015; Ryabenko et al., 2012), which preferentially uses the light isotope and leaves the residual nitrate enriched in $^{15}$N. A notable difference between the EANN and a previous biogeochemical model estimate of nitrate $\delta^{15}$N (Somes et al.,
2010) is that the EANN captures the higher nitrate $\delta^{15}$N in the Arabian Sea compared to the Bay of Bengal (a relation-

ship that is expected based on active denitrification in the Arabian Sea).

Lowest $\delta^{15}$N values of subsurface nitrate are (250 m) found in the Southern Ocean and in the North Atlantic. The
55 North Atlantic subtropical gyre in particular has the lowest $\delta^{15}$N values in any basin (Fig. 5b; also see Marconi et al., 2015, 2017; Fawcett et al., 2011; Knapp et al., 2005, 2008), which has been attributed to the remineralization of low-$\delta^{15}$N organic matter originating from $N_2$ fixation, which pro-
60 duces organic matter with a $\delta^{15}$N between 0 and $-1$‰ (similar to atmospheric $N_2$; see Fig. 1b; Carpenter et al., 1997; Hoering and Ford, 1960). Earlier work argues that this nitrate $\delta^{15}$N lowering requires the bulk of Atlantic $N_2$ fixation ($\approx 90$ %) to occur in the tropics (Marconi et al., 2017) fol-
65 lowed by the advection of this remineralized nitrate to the North Atlantic. This contrasts with numerical models arguing for high $N_2$ fixation rates in the North Atlantic (Ko et al., 2018). Similar local minima of subsurface $\delta^{15}$N appear in all the subtropical gyres (Fig. 5b), which is consistent with
70 observations (Casciotti et al., 2008; Yoshikawa et al., 2015) and presumably indicates the importance of $N_2$ fixation in these regions (e.g., Ko et al., 2018) and others. The $N_2$ fixation $\delta^{15}$N signal in the Pacific Ocean (lowering in situ nitrate $\delta^{15}$N via remineralization of organic matter with lower
$\delta^{15}$N) will be counteracted by the influence of water column denitrification in that basin (raising the $\delta^{15}$N of residual nitrate), which imparts a high-$\delta^{15}$N signal, but a local minimum in $\delta^{15}$N can still be seen in the Pacific subtropical gyres (Fig. 4a).

Nitrate assimilation by phytoplankton in the upper ocean is influenced by both the subsurface source nitrate $\delta^{15}$N and the degree of nitrate assimilation (Miyake and Wada, 1967; Wada and Hattori, 1978) (Fig. 1b). This gives the expectation that average nitrate $\delta^{15}$N values for the upper 50 m (Fig. 5a)
85 will be consistently higher than those at 250 m (Fig. 5b). However, the highest observed and estimated values in the upper 50 m are not found above the ODZ regions, but are on the edges of high nitrate concentration upwelling zones in the Southern Ocean, equatorial Pacific, and subarctic gyres (con-
90 tours in Fig. 2). Circulation in these "edge" regions allows for nitrate to be advected along the surface, lengthening its time in the surface ocean and allowing more utilization to elevate the residual nitrate $\delta^{15}$N pool. In other words, the degree of nitrate utilization appears to play a more important role in
determining surface nitrate $\delta^{15}$N than the initial value. (This is not the case for the organic matter $\delta^{15}$N produced from this nitrate, which will be discussed more below.)

Despite our expectation of higher nitrate $\delta^{15}$N in the upper 50 m versus 250 m (Fig. 5a vs. b), we identify two types of
100 regions where this difference is negative (Fig. 6): above ODZ waters and in subtropical gyres. The explanation for the negative values above the ODZ regions is that the nitrate $\delta^{15}$N at 250 m must be much higher than the nitrate $\delta^{15}$N upwelled to the surface. This is consistent with elevated ODZ nitrate
$\delta^{15}$N having an indirect path to waters outside of ODZ re-

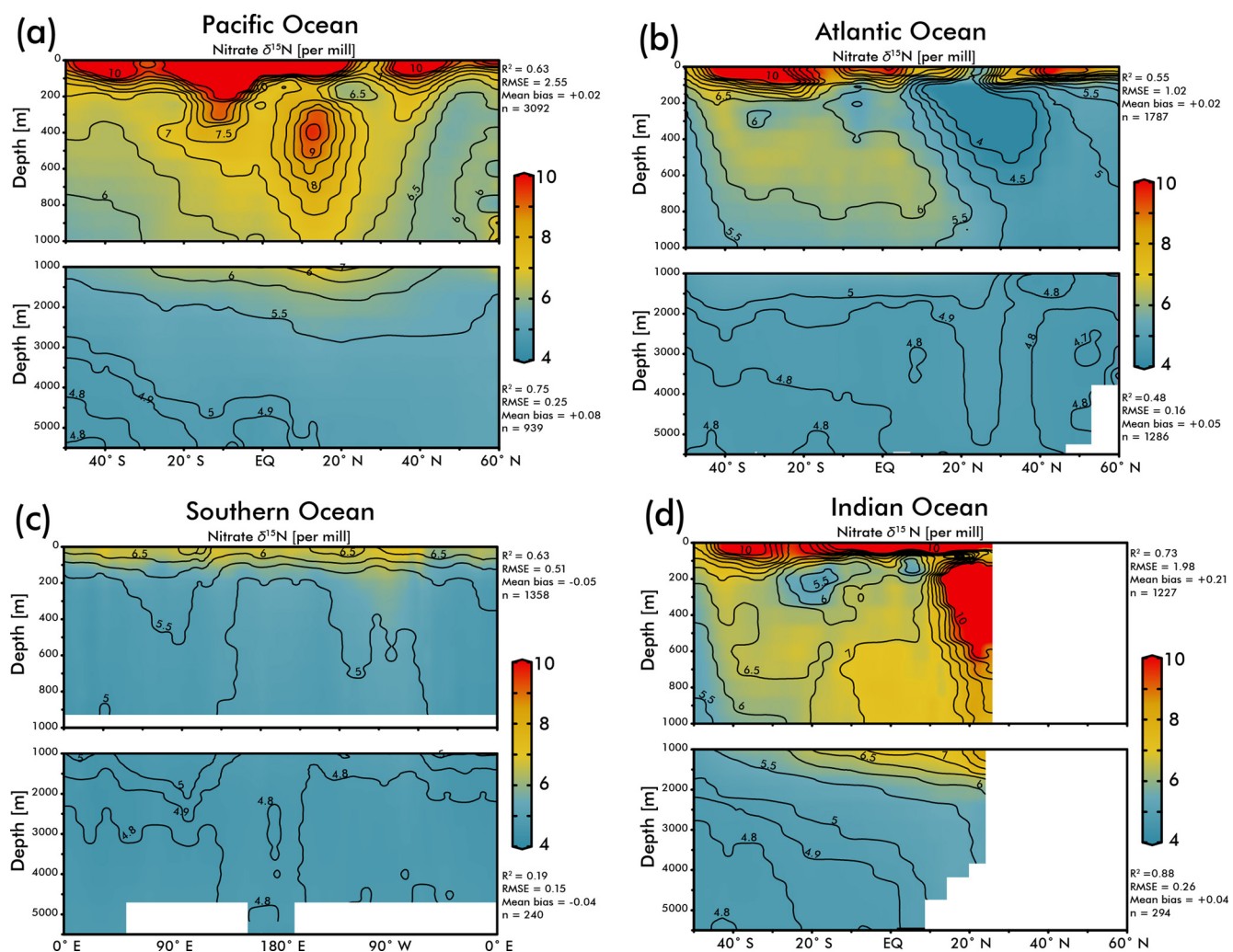

**Figure 4.** Zonal average of model nitrate $\delta^{15}$N versus latitude or longitude for the **(a)** Pacific Ocean, **(b)** Atlantic Ocean, **(c)** Southern Ocean, and the **(d)** Indian Ocean. White bars indicate no data because of land. The $R^2$, RMSE, mean bias, and total number ($n$) of observed versus EANN nitrate $\delta^{15}$N are shown on the right for each region and depth range. White indicates regions of no data coverage. Note that these zonally averaged views obscure zonal gradients in nitrate $\delta^{15}$N (see Fig. 5).

gions (Peters et al., 2017; Rafter et al., 2013). The subtropical gyres also have modeled nitrate $\delta^{15}$N in the upper 50 m that is less than 250 m, but this finding is difficult to test with observations because of low nitrate concentrations. That said,
the model predicts a lower nitrate $\delta^{15}$N in the upper ocean relative to that at 250 m, which is consistent with $N_2$ fixation in these regions.

Our discussion above highlights the difficulty of distinguishing between the competing influences of the subsurface
source nitrate $\delta^{15}$N and the degree of nitrate utilization on residual nitrate $\delta^{15}$N. Clearly a static depth does not reflect the subsurface source of nitrate delivered to the surface and a more robust method for estimating this subsurface source needs to be developed. However, some generalizations can
be made regarding the organic matter $\delta^{15}$N produced in these regions and its potential influence (via remineralization) on

subsurface nitrate throughout the water column via the export and remineralization of organic matter (Sigman et al., 2009a). For example, a local minimum in $\delta^{15}$N is visible at 250 m depth in the eastern equatorial Pacific (Fig. 5b; also   20
discussed in several studies; Rafter et al., 2012; Rafter and Sigman, 2016) is caused by the remineralization of organic matter with a low $\delta^{15}$N due to partial nitrate consumption at the surface. Below we discuss these and other influences on intermediate-depth nitrate $\delta^{15}$N.   25

## 4.2   Intermediate-depth nitrate $\delta^{15}$N variability

Waters at "intermediate" depths (here shown as the 750 m surface in Fig. 5c) are important because they are part of a large-scale circulation that originally upwells in the Southern Ocean and resupplies nutrients to the low-latitude ther-   30

**Figure 5.** (Left) Map view of nitrate $\delta^{15}$N from our EANN and our observations (circles) for the **(a)** average over the 0–50 m depth as well as the **(b)** 250 m, **(c)** 700 m, and **(d)** 3000 m depth surfaces. (Right) Map views of nitrate $\delta^{15}$N error from the EANN model nitrate $\delta^{15}$N for the same depth surfaces on the left.

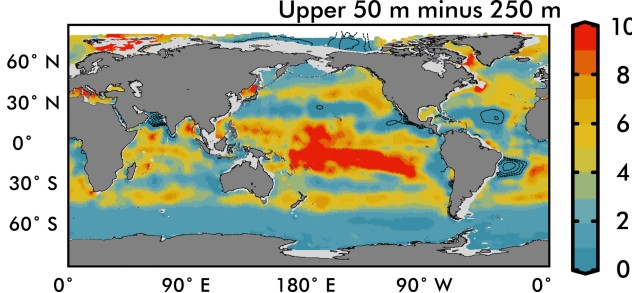

**Figure 6.** Difference between the average nitrate $\delta^{15}$N in the upper 50 m and 250 m depths in Fig. 5. Dashed contours in low-latitude ODZ regions and subtropical gyres indicate regions where nitrate $\delta^{15}$N at 250 m is greater than the upper 50 m nitrate $\delta^{15}$N.

mocline (Palter et al., 2010; Sarmiento et al., 2004; Toggweiler et al., 1991; Toggweiler and Carson, 1995). Within the context of this overturning, the nitrate $\delta^{15}$N upwelling in the Southern Ocean begins at $\approx 5\,‰$ (Figs. 4c and 5c) and is elevated $\approx 1\,‰$ by partial nitrate assimilation in the Southern Ocean surface waters as they are advected equatorward (see Figs. 5a and 6; Rafter et al., 2012). Deep wintertime mixing in the subantarctic Pacific converts these Southern Ocean surface waters into mode and intermediate waters (Herraiz-Borreguero and Rintoul, 2011), introducing nitrate with a "preformed" $\delta^{15}$N of $\approx 6\,‰$ into the intermediate-depth South Pacific and South Atlantic (Rafter et al., 2012, 2013; Tuerena et al., 2015) at depths between $\approx 600$ and 1200 m. TSI The penetration of this preformed signal (nitrate $\geq 6\,‰$) into the interior can be clearly seen in the Atlantic Ocean between $\approx 40°$ S and 20° N (Fig. 4b).

The TS2 same signal is carried with Southern Ocean mode and intermediate waters into the Pacific basin as far as the tropics (Lehmann et al., 2018; Rafter et al., 2013), although it is difficult to distinguish in the model results against the higher background $\delta^{15}N$ in this basins (Figs. 4a, d, 5c). The same process presumably introduces elevated nitrate $\delta^{15}N$ to the Indian Ocean, which has similar values at this depth. Nitrate $\delta^{15}N$ increases from the Southern Ocean toward the Equator in the Pacific Ocean and the Indian Ocean, but not in the Atlantic Ocean (Fig. 5). Organic matter TS3 has a lower $\delta^{15}N$ in the Atlantic Ocean than in the Pacific and Indian oceans because of a lack of water column denitrification supplying high-$\delta^{15}N$ water to the surface and because of the high rates of $N_2$ fixation which supply isotopically light N to organic matter (Marconi et al., 2017; Tuerena et al., 2015). This contrast in intermediate-depth nitrate $\delta^{15}N$ is likely caused by differences in $\delta^{15}N$ of organic matter remineralized in these regions region. The increase in intermediate-depth nitrate $\delta^{15}N$ from the subantarctic to the tropical Pacific appears to result from the remineralization of organic matter with a $\delta^{15}N$ elevated by elevated source nitrate $\delta^{15}N$ (TS4 near the ODZ) or extreme elevation of residual nitrate $\delta^{15}N$ (advected along the surface away from the Equator; see high surface nitrate $\delta^{15}N$ in Fig. 5a). Previous work suggests that direct mixing with denitrified waters represents only a small fraction of the change from the preformed high-latitude value ($\approx 6\permil$) to tropical nitrate $\delta^{15}N$ of $\approx 7\permil$ (Peters et al., 2017; Rafter et al., 2012, 2013).

The southern Indian Ocean is one region particularly devoid of published nitrate $\delta^{15}N$ observations (Fig. 2), but the EANN makes specific predictions about its distribution. For example, the modeled nitrate $\delta^{15}N$ predicts that intermediate-depth Indian Ocean nitrate is elevated in $\delta^{15}N$ similarly to the intermediate-depth South Pacific (Fig. 5c). Considering that both intermediate-depth water masses are formed from Southern Ocean surface waters, it is reasonable to propose that nitrate $\delta^{15}N$ are similarly elevated by partial nitrate consumption. The EANN therefore provides testable predictions for nitrate $\delta^{15}N$ observations throughout the Indian Ocean.

## 4.3 Deep-sea nitrate $\delta^{15}N$ trends

Our discussion above suggests that the basin-scale balance of $N_2$ fixation and water column denitrification is a major contributor to inter-basin nitrate $\delta^{15}N$ gradients in the upper ocean ($< 1000$ m; see Fig. 5), lowering values in the Atlantic Ocean compared to the Pacific and Indian oceans. By averaging EANN nitrate $\delta^{15}N$ with depth for each ocean basin (Fig. 7), we find that these basin-scale nitrate $\delta^{15}N$ differences also persist into the deep sea (here defined as $\geq 3000$ m and below). Note that the inter-basin EANN nitrate $\delta^{15}N$ gradients in Fig. 7a are smaller than the corresponding inter-basin gradients in observed $\delta^{15}N$, because the observations are spatially biased towards areas of water column denitrification in the Pacific and Indian oceans (see Fig. 2).

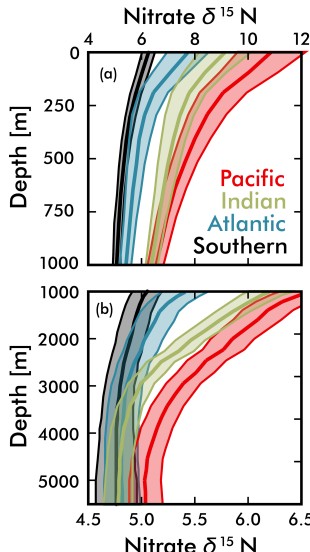

**Figure 7.** Mean EANN nitrate $\delta^{15}N$ (solid line) and $1\sigma$ standard deviation (envelope) with depth for each ocean basin. Note change in vertical and horizontal axes between **(a)** and **(b)**.

The remineralization of organic matter is one process that can – and has been used to – explain both the elevation of deep Pacific nitrate $\delta^{15}N$ (Peters et al., 2017; Rafter et al., 2013; Sigman et al., 2009a) and lowering of deep Atlantic nitrate $\delta^{15}N$ (Knapp et al., 2008; Marconi et al., 2017; Tuerena et al., 2015) relative to the deep ocean mean. Here we provide two additional pieces of evidence that argue for the remineralization of organic matter as the key driver of these deep-sea nitrate $\delta^{15}N$ differences. Our first piece of evidence is that the average subsurface source of nitrate to the Pacific Ocean and Indian Ocean surfaces has a significantly higher $\delta^{15}N$ (by $2\permil$ at the 250 m depth surface) than the Atlantic and Southern Ocean (Figs. 5b and 7). Nitrate $\delta^{15}N$ at 250 m is an admittedly imprecise estimate for the nitrate upwelled to the surface, but even a slight elevation in Pacific source nitrate $\delta^{15}N$ and near-complete nitrate utilization at the surface will translate into higher sinking organic matter $\delta^{15}N$ (i.e., see Fig. 1b).

Our second piece of evidence that the export and remineralization of organic matter drives the inter-basin differences in deep nitrate $\delta^{15}N$ comes from sediment trap measurements. Averaging published sediment trap organic matter $\delta^{15}N$ from the subtropical and tropical Pacific gives a value of $8.5 \pm 2.9\permil$ (Knapp et al., 2016; Robinson et al., 2012), which is significantly higher than measured in traps from the Atlantic ($4.5 \pm 1.5\permil$) (Freudenthal et al., 2001; Holmes et al., 2002; Lavik, 2000; Thunell et al., 2004). Given observed Southern Ocean nitrate characteristics (Rafter et al., 2013), we estimate an even lower typical sinking organic matter $\delta^{15}N$ of $+1.5\permil$ for this region, which assumes initial nitrate values equal the upper circumpolar deep water and final values from the open Antarctic zone. This value is con-

sistent with annually averaged sinking organic matter $\delta^{15}N$ of $\approx 0.9\,‰$ to $1.6\,‰$ in the open Antarctic zone (Lourey et al., 2003), although published results from the iron-fertilized Kerguelen Plateau region are predictably higher (Trull et al., 2008). The much lower Southern Ocean sinking organic matter $\delta^{15}N$ is consistent with partial consumption of nitrate at the surface (see Fig. 1b), and the entrainment of this nitrate in equatorward-moving intermediate waters acts to export nitrate with elevated $\delta^{15}N$ to intermediate waters throughout the Southern Hemisphere (see discussion above and Rafter et al., 2012, 2013). Based on this evidence, it appears that global patterns of sinking organic matter $\delta^{15}N$ are consistent with the remineralization of this organic matter driving subtle, but significant differences in deep-sea nitrate $\delta^{15}N$.

An alternative explanation for the deep-sea nitrate $\delta^{15}N$ differences in Fig. 7 and Table 1 is that they reflect the lateral (along isopycnal) advection of elevated nitrate $\delta^{15}N$ from the shallower Pacific and Indian Ocean ODZ regions. However, we can easily dismiss this explanation by looking at the meridional trends in deep-sea nitrate $\delta^{15}N$ – following the deep waters from their entrance in the south and movement northward. What we find is that deep EANN nitrate $\delta^{15}N$ (Fig. 5d) is lowest in the Southern Ocean and increases equatorward in the Pacific (Table 1). Average deep-sea nitrate $\delta^{15}N$ for the global ocean is $5.0 \pm 0.3\,‰$, but average observed nitrate $\delta^{15}N$ below 3000 m increases from $4.7 \pm 0.1\,‰$ in the Pacific sector of the Southern Ocean to $4.9 \pm 0.2\,‰$ in the deep South Pacific, $5.4 \pm 0.2\,‰$ in the deep tropical Pacific, and $5.2 \pm 0.2\,‰$ in the deep North Pacific (Table 1). This is consistent elevated organic matter $\delta^{15}N$ being produced and exported from the lower-latitude Pacific Ocean surface being remineralized at depth. This is also consistent with the known increase in nitrate concentrations and lowering of deep oxygen concentrations from the deep South Pacific to tropical and North Pacific (e.g., see Fig. 4e in Rafter et al., 2013). This contrasts with the Atlantic, where there is with no significant change in deep nitrate $\delta^{15}N$, despite the export of slightly elevated nitrate $\delta^{15}N$ into intermediate-depth Atlantic (see above and Tuerena et al., 2015). In other words, inter-basin differences in sinking organic matter $\delta^{15}N$ best explains the inter-basin differences in deep EANN and observed nitrate $\delta^{15}N$. Diapycnal mixing from the low-latitude Pacific ODZ regions may also play a part in the south-to-north elevation of deep Pacific nitrate $\delta^{15}N$ – a contribution that can be quantified by future work with a circulation model.

**Table 1.** Average EANN nitrate $\delta^{15}N \geq 3000$ m for each ocean region (tropical being between 23.5° N and 23.5° S).

|          | Indian | Pacific | Atlantic |
|----------|--------|---------|----------|
| North    | –      | $5.4 \pm 0.2$ | $4.8 \pm 0.1$ |
| Tropical | $5.1 \pm 0.2$ | $5.2 \pm 0.2$ | $4.9 \pm 0.1$ |
| South    | $4.8 \pm 0.1$ | $5.0 \pm 0.2$ | $4.8 \pm 0.1$ |

|  | Southern | Global |
|--|----------|--------|
|  | $4.8 \pm 0.1$ | $5.0 \pm 0.3$ |

## 5   Conclusions

We find that an ensemble of artificial neural networks (EANN) can be trained on climatological distributions of physical and biogeochemical tracers to reproduce a global database of nitrate $\delta^{15}N$ observations (Fig. 2) with good fidelity (Fig. 3). We used the EANN to produce global climatological maps of nitrate $\delta^{15}N$ at a 1° resolution from the surface to the seafloor. These results help identify spatial patterns (Figs. 4–6) and quantify regional and basin-average oceanic values of nitrate $\delta^{15}N$ (Fig. 7 and Table 1). Major differences between the observed and EANN-predicted nitrate $\delta^{15}N$ appear to be caused by temporal variability of nitrate $\delta^{15}N$ in the upper ocean and in ODZs associated with variable nitrate uptake and denitrification rates. Additional measurements of nitrate $\delta^{15}N$ will help to develop seasonally resolved maps that can improve upon the climatological mean map provided here.

*Code and data availability.* Observations can be obtained at https://www.bco-dmo.org/dataset/768627 (last access: 14 June 2019). Model output and code can be obtained at https://www.bco-dmo.org/dataset/768655 (last access: 14 June 2019). TS5

## Appendix A

**Table A1.** Contributing analysts and references for citing original data sources.

| Region | Analyst | Institution | Citation |
| --- | --- | --- | --- |
| Pacific North – Subarctic | Mark Altabet | U of Massachusetts, Dartmouth | Altabet and Francois (1994b) |
| Pacific North – Subarctic | Eitaro Wada | JAMSTEC | Wada (1980) |
| Pacific North – Subarctic | Eric Galbraith | McGill University | Galbraith (2006) |
| Pacific North – Subarctic | Jinping Wu | U of British Columbia | Wu et al. (1997) |
| Pacific North – Subarctic | Moritz Lehmann | University of Basel | Lehmann et al. (2005) |
| Pacific North – Bering Sea | Julie Granger | U of Connecticut | Granger et al. (2011, 2013) |
| Pacific North – Okhotsk | Chisato Yoshikawa | JAMSTEC | Yoshikawa et al. (2006) |
| Pacific North – Kuroshio | Kon-Kee Liu (deceased) | National Central University, Taiwan | Liu et al. (1996) |
| Pacific North – South China Sea | George Wong | Old Dominion University | Wong et al. (2002) |
| Pacific North | Julie Granger | U of Connecticut | J. Granger (unpublished data) |
| Pacific North | Julie Granger | U of Connecticut | Umezawa et al. (2014) |
| Pacific North – Gulf of California | Mark Altabet | U of Massachusetts, Dartmouth | Altabet et al. (1999) |
| Pacific North – Tropical | Daniel Sigman | Princeton University | Sigman et al. (2005) |
| Pacific North – ALOHA | Daniel Sigman | Princeton University | Sigman et al. (2009b) |
| Pacific North – ALOHA | Karen Casciotti | Stanford University | Casciotti et al. (2008) |
| Pacific North | Angela Knapp | Florida State University | Knapp et al. (2011) |
| Pacific North – Tropical | Jay Brandes | Skidaway Institute of Oceanography | Brandes et al. (1998) |
| Pacific North – Tropical | Maren Voß (Voss) | Leibniz Institute for Baltic Sea Research | Voss et al. (2001) |
| Pacific North – Tropical | Moritz Lehmann | University of Basel | Kienast et al. (2008) |
| Pacific North – Tropical | Nadine Lehmann | Dalhousie University | Lehmann et al. (2018) |
| Pacific North – Tropical | Karen Casciotti | Stanford University | Casciotti and McIlvin (2007) |
| Pacific – Tropical | Patrick Rafter | UC Irvine | Rafter et al. (2012), Rafter and Sigman (2016) |
| Pacific South – Tropical | Kon-Kee Liu (deceased) | National Central University, Taiwan | Liu (1979) |
| Pacific South – Tropical | Ricardo de Pol-Holz | Centro de Investigación GAIA-Antártica (CIGA) | De Pol-Holz et al. (2009) |
| Pacific South – Tropical | Evgeniya Ryabenko | GEOMAR | Ryabenko et al. (2012) |
| Pacific South – Tropical | Brian Peters | Stanford University | Peters et al. (2017) |
| Pacific South – Tropical | Angela Knapp | Florida State University | Knapp et al. (2016) |
| Pacific South – Tropical | Karen Casciotti | Stanford University | Casciotti et al. (2013) |
| Pacific South – Tropical | Annie Bourbonnais | University of South Carolina | Bourbonnais et al. (2015) |
| Pacific South | Chisato Yoshikawa | JAMSTEC | Yoshikawa et al. (2015) |
| Pacific South – Tasman Sea | Patrick Rafter | UC Irvine | P. Rafter and D. Sigman [TS6] (unpublished data) |
| Pacific South – Tropical | Patrick Rafter | UC Irvine | Rafter et al. (2012) |
| Pacific South – Chilean Margin | Patrick Rafter | UC Irvine | Rafter et al. (2013) |
| Southern Ocean – Pacific | Mark Altabet | U of Massachusetts, Dartmouth | Altabet and Francois (2001) |
| Southern Ocean – Pacific | Daniel Sigman | Princeton University | Sigman et al. (1999a) |
| Southern Ocean – Pacific | Preston Kemeny | Cal Tech | Kemeny et al. (2016) |
| Southern Ocean – Indian | Karen Casciotti | Stanford University | Trull et al. (2008) |
| Southern Ocean – Indian | Daniel Sigman | Princeton University | Sigman et al. (1999a) |
| Southern Ocean – Indian | Peter DiFiore | Princeton University | DiFiore et al. (2006) |
| Southern Ocean – Atlantic | Sandi Smart | Stellenbosch University | Smart et al. (2015) |
| Atlantic North – Subarctic | Patrick Rafter | UC Irvine | P. Rafter and D. Sigman [TS7] (unpublished data) |
| Atlantic North – Subarctic | Dario Marconi | Princeton University | Marconi et al. (2017) |
| Atlantic North – Subarctic | Maren Voß (Voss) | Leibniz Institute for Baltic Sea Research | Voss (1991) |
| Atlantic North | Annie Bourbonnais | University of South Carolina | Bourbonnais et al. (2009) |
| Atlantic North | Karen Casciotti | Stanford University | Schlitzer et al. (2018) |
| Atlantic North | Angela Knapp | Florida State University | Knapp et al. (2005) |
| Atlantic North | Angela Knapp | Florida State University | Knapp et al. (2008) |
| Atlantic North | Dario Marconi | Princeton University | Marconi et al. (2015) |

| Region | Analyst | Institution | Citation |
| --- | --- | --- | --- |
| Atlantic Tropical | Martin Frank | IFM-Geomar | Schlitzer et al. (2018) |
| Atlantic South | Robyn Tuerena | University of Edinburgh | Tuerena et al. (2015) |
| Mediterranean | Julian Sachs | University of Washington | Sachs and Repeta TS8 (1999) |
| Mediterranean | Silvio Pantoja | University of Concepción | Pantoja et al. (2002) |
| Indian North | Birgit Gaye | Universität Hamburg | Gaye et al. (2013) |
| Indian – Arabian Sea | Mark Altabet | U of Massachusetts, Dartmouth | Altabet et al. (1999) |
| Indian – Arabian Sea | Jay Brandes | Skidaway Institute of Oceanography | Brandes et al. (1998) |
| Indian – Arabian Sea | Taylor Martin | Stanford University | Martin and Casciotti (2017) |
| Indian – Arabian Sea | Patrick Rafter | UC Irvine | DeVries et al. (2013) |
| Indian South | Kristen Karsh | CSIRO | Karsh et al. (2003) |
| Indian South | Frank Dehairs | Vrije Universiteit Brussel | Dehairs et al. (2015) |
| Arctic Ocean | Julie Granger | U of Connecticut | J. Granger (unpublished data) |
| Arctic Ocean | Francois Fripiat | Université Libre de Bruxelles | Fripiat et al. (2018) |

*Author contributions.* Data compilation was done by PAR. Coding and statistics were done by AB, TD, and PAR. All authors contributed to writing the manuscript.

*Competing interests.* The authors declare that they have no conflict of interest.

*Acknowledgements.* We thank Mark Altabet, Karen Casciotti, Alyson Santoro, Benoit Pasquier, J. J. Becker, two anonymous reviewers, and Markus Kienast, as well as Julie Granger and Daniel Sigman for (at-the-time) unpublished data. A complete list of references can be found in the Appendix. Many figures were made using Ocean Data View software (Schlitzer, 2002). Custom-made color palettes are available via https://prafter.com/ (last access: 1 January 2019).

*Review statement.* This paper was edited by Perran Cook and reviewed by two anonymous referees.

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

**Remarks from the language copy-editor**

CE1    Please reconsider the requested deletion as it does not make sense with regards to the sentence structure.

CE2    Please note the slight edits made to your request

**Remarks from the typesetter**

TS1    Please note that the requested changes here would have to be approved by the editor. If these changes should still be made, please provide a short explanation regarding these corrections that can be forwarded by us to the editor.

TS2    Please see previous remark.

TS3    Please see previous remark.

TS4    Changes here and in following paragraphs would need to be approved by the editor.

TS5    If possible, please provide references to the data set and the model output mentioning author list, title, URL, last access date and year of upload.

TS6    Please confirm.

TS7    Please confirm.

TS8    Please confirm.

TS9    Please provide all author names.

TS10    If there are too many author names, it is our house standard to only use three names from the author list