# Peer review of "Global trends in marine nitrate N isotopes from observations and a neural network-based climatology"

_Biogeosciences, 2018_

## Referee Comment (RC1) · Anonymous Referee #1 · 1 Feb 2019

Overview: The paper targets a useful goal – providing a map of 15N-NO3 estimates for the global ocean for use in biogeochemical studies. To do this, it uses a neural network to obtain a relationship between sparse observed 15N-NO3 and World Ocean Atlas (WOA) values of temperature, salinity, oxygen, phosphate and nitrate, and then maps the derived 15N-NO3 estimates. The utility of the approach is assessed via correlation statistics between the estimates and the observations. There are areas where the estimates and observations agree well and others where they agree poorly. The latter are ascribed to temporal offsets between the WOA data collection and the 15N-NO3 observations. As far as it goes, the paper is sound, but it doesn't go very far (as an aside it does provide clear and well-constructed descriptions of possible mechanistic

causes of the spatial variations in the 15N-NO3 observations, although these do not really derive from or depend on the mapping exercise). It could be improved by addressing the following issues: 1. Is the neural network (NN) approach demonstrably better than a multiple linear regression (MLR) to the same input variables? Assessing this would be useful for two reasons: a. The MLR has the advantage that is provides a simple equation that all can use with their local and future input variable observations [(notably MLR approaches are becoming widely used for nitrate in the context of BGC-Argo observations; Carter et al. 2017, https://doi.org/10.1002/lom3.10232] b. Determining whether and in which parts of the ocean the non-linear NN approach outperforms the linear MLR approach is likely to shed light on the processes that drive 15N-NO3 variations. 2. Are there other metrics that could assess possible causes of the quality of the matches and mismatches between estimates and observations, to go beyond simply ascribing them to temporal offsets? For example since some of the 15N-NO3 estimates were probably collected synchronously with the WOA data, do these points show closer agreement? Can agreement with mechanistic understanding be assessed – for example in regions where single processes largely dominate 15N-NO3 variations (e.g. nitrate assimilation in Southern Ocean surface waters) does the NN approach produce sensible correlations between [nitrate] and 15N-NO3 ?

Details: Line 63: ammonia assimilation is also a significant determinant of the 15N of organic matter. Line 370: meaning of sentence beginning "Equivalent processes... was opaque to me. Lines 384-395: This discussion of separating nitrification from denitrification influences on deep water 15N-NO3 values would benefit from recognition that relationships with O2 and nitrate have opposite signs. Line 403: The estimate low sinking organic matter 5N estimate of +1.5 should be compared to published results in Lourey et al., 2003, GBC, which show good agreement: https://doi.org/10.1029/2002GB001973
* * *

---

## Referee Comment (RC2) · Anonymous Referee #2 · 16 Feb 2019

The nitrate isotope database and gridded product generated by the authors has the potential to be extremely valuable for studies of the marine nitrogen cycle. I commend them for undertaking this important task, which will benefit researchers broadly. Because it does have such strong potential utility, I would really like to see the paper describe a bit more clearly what was actually done here, and how it compares with other methods of data gridding.

In particular, I think the authors should further explain and reference the neural network model used to generate the gridded product. There's only one paper in the references, from 1996, that seems to relate at all to the methods they applied. More detail should

be given here so that the results could be reproduced, or extended as additional nitrate isotope data become available.

Next, the discussion and conclusions about the marine nitrogen cycle were largely confirmatory of earlier studies, but also almost beside the point of this particular manuscript. I would have found it more interesting, in the context of what was done here, to see how this kind of approach to data binning compares to alternative methods. Are there significant difference between this neural network approach, and a World Ocean Atlas approach of data interpolation? What are the implications of some of the choices made in building the model?

Specific comments are given below.

Lines 106-111: How does this neural network actually work? Does it use learning based on surrounding data to inform the values of unknown points? Where are the equations that go into the model? What is/are the function(s) that produces $\delta$15N values from the gridded T, S, NO3-, O2, and PO43- data?

Lines 116-119: Please clarify the description of depth binning.

Lines 122-123: Why were whole ship tracks used for validation, rather than a more random selection?

Line 131: How was the daily chlorophyll used in an otherwise annual gridded product?

Section 2.2 needs more references, especially 2.2.3 (lines 137-151). There is a lot of terminology here that is not defined or referenced, such as hidden layer, node, activation function, which should be defined and explained further. Also, it is not clear what you are applying weights to in the model.

Line 158: It says that 10% of the observations were withheld to validate the networks. How were these chosen? More generally, how were the data for training, text, and validation chosen?

Line 165: What are the implications of using whole cruise tracks for external validation rather than randomly chosen stations or grid cells?

Lines 179-180: Could this be shown (that the ensemble performs better than any single member of the ensemble) using your results, or is this a general feature? Does it apply here?

Discussion section:

How does the discussion stem from their results from the neural network model? Most of the discussion seems to focus on general features discussed in the original papers about the data used to generate the product. It would be more satisfying for this reviewer to read about how some of the choices they made in producing the model impacted the results.

For example, 1) Is there only one way to produce the neural network model? 2) How were choices made? What tradeoffs were tolerated? What are the implications? 3) How does this approach compare with other methods for gridding? 4) Are there particular nodes that performed well in some locations vs. others?

Lines 415-423: It's not clear how the authors 'easily dismiss' an explanation about lateral advection of elevated nitrate $\delta$15N from ODZ regions. I think this section should be clarified. The way they set it up (seeing an increase in the Pacific but not Atlantic) does not seem to further the argument they are trying to make since the largest ODZ regions are in the Pacific, not the Atlantic.

Figures— Figure 2—How many different different selections of training, test, and validation sets did the authors test in the neural network model? What was the rationale behind the choice of the whole cruise tracks that were used for validation?

Figure 3—Panel C was helpful. Panels A and B were also useful, but the choice of the non-linear color scale bar, where most of the data points were off scale, was unusual. In panel A, also please clarify whether this includes all of the data, or just those form

the training set? Or validation set?

Figure 4—The statistics for the different zonally averaged sections were useful, but I question the utility of the zonally averaged Pacific, given some of the large zonal gradients in $\delta$15N from the ODZs in the eastern tropical Pacific.

Figure 5—The contours were extremely difficult to read, and the panels on the right hand side (E-H) were not particularly helpful. I also wondered how much of the patchiness, especially in panel A, is driven by the distribution of available $\delta$15N data?

Figure 6—Again, the contours are difficult to see. Can you indicate negative numbers with a different color, or allow the color bar to include negative numbers?

---

## Author Comment (AC1) · 7 Mar 2019

**The original comment is in bold font**. The response to the comment is in regular font.

**Overview: The paper targets a useful goal – providing a map of 15N-NO3 estimates for the global ocean for use in biogeochemical studies. To do this, it uses a neural network to obtain a relationship between sparse observed 15N-NO3 and World Ocean Atlas (WOA) values of temperature, salinity, oxygen, phosphate and nitrate, and then maps the derived 15N-NO3 estimates. The utility of the approach is assessed via correlation statistics between the estimates and the observations. There are areas where the estimates and observations agree well and others where they agree poorly. The latter are ascribed to temporal offsets between the WOA data collection and the 15N-NO3 observations.**

To be clear, our interpretation of the observation-model comparison is that the model estimates the mean values quite well, but does not include temporal variability and therefore will not capture temporal variability.

**As far as it goes, the paper is sound, but it doesn't go very far (as an aside it does provide clear and well-constructed descriptions of possible mechanistic causes of the spatial variations in the 15N-NO3 observations, although these do not really derive from or depend on the mapping exercise). It could be improved by addressing the following issues: 1. Is the neural network (NN) approach demonstrably better than a multiple linear regression (MLR) to the same input variables? Assessing this would be useful for two reasons: a. The MLR has the advantage that is provides a simple equation that all can use with their local and future input variable observations [(notably MLR approaches are becoming widely used for nitrate in the context of BGC-Argo observations; Carter et al. 2017, https://doi.org/10.1002/lom3.10232] b. Determining whether and in which parts of the ocean the non-linear NN approach out- performs the linear MLR approach is likely to shed light on the processes that drive 15N-NO3 variations.**

Great comment. To address this we built a single global Multiple Linear Regression (MLR) model using all the same predictors used in the Ensemble Array of Neural Networks (EANN). We found that the MLR performs much worse than the EANN at predicting nitrate δ15N. The coefficient of determination for each method and each ocean basin's upper 1000 m is shown in the table below.

| | Atlantic | Pacific | Indian | Southern Ocean |
|---|---|---|---|---|
| MLR R2 | 0.04 | 0.49 | 0.51 | 0.34 |
| EANN R2 | 0.53 | 0.78 | 0.76 | 0.68 |

The reason for this worse performance is likely that the MLR approach assumes the training parameters are independent of each other, but also dependent on nitrate δ15N. This is not the case and so the EANN approach performs noticeably better.

**2.  Are there other metrics that could assess possible causes of the quality of the matches and mismatches between estimates and observations, to go beyond simply ascribing them to temporal offsets? For example since some of the 15N-NO3 estimates were probably collected synchronously with the WOA data, do these points show closer agreement?**
We do not ascribe differences between model and observations to temporal offsets. We suggest that the model predicts an annual climatology of nitrate δ15N, while the observations measure the instantaneous δ15N. There is no temporal component in the EANN. The WOA data that we are using are the annual climatologies – there are no corresponding observations of δ15N.

**Can agreement with mechanistic understanding be assessed – for example in regions where single processes largely dominate 15N- NO3 variations (e.g. nitrate assimilation in Southern Ocean surface waters) does the NN approach produce sensible correlations between [nitrate] and 15N-NO3 ?**

This is a good suggestion, but we find that adding an additional analysis of the regional model estimates is beyond the scope of this paper. In fact, we are already using the EANN results to examine global nitrate uptake patterns in a current study that will be outlined in a dedicated manuscript.

**Details: Line 63: ammonia assimilation is also a significant determinant of the 15N of organic matter.**
We revised the manuscript to clarify that these sentences refer to organic matter production by the assimilation of nitrate. Good comment.

**Line 370: meaning of sentence beginning "Equivalent processes… was opaque.**
The revised manuscript clarifies this sentence. It refers to how the model nitrate $\delta^{15}$N predicts that intermediate water nitrate $\delta^{15}$N in the Indian Ocean has a similar value as the corresponding waters in the Pacific. We argue that this is likely because "equivalent processes" established the pre-formed characteristics of both water masses (i.e., partial nitrate assimilation in the Southern Ocean surface).

**Lines 384-395: This discussion of separating nitrification from denitrification influences on deep water 15N-NO3 values would benefit from recognition that relationships with O2 and nitrate have opposite signs.**
Good comment. The well-known south-to-north lowering of deep Pacific O2 and increase in nitrate concentrations is consistent with the remineralization of organic matter and not the lateral advection of nitrate from ODZ regions. This will be added to the revised manuscript.

**Line 403: The estimate low sinking organic matter d15N estimate of +1.5 should be compared to published results in Lourey et al., 2003, which show good agreement.**
We have added and refer to this citation's results in the revised manuscript.

---

## Author Comment (AC2) · 7 Mar 2019

There are many detailed responses to Reviewer #2's comments. We have stated where these responses translate into revised text in the manuscript. Please let us know if there are any comments that should also drive a revision of manuscript text.

**The original comment is in bold font**. The response to the comment is in regular font.

**The nitrate isotope database and gridded product generated by the authors has the potential to be extremely valuable for studies of the marine nitrogen cycle. I commend them for undertaking this important task, which will benefit researchers broadly. Because it does have such strong potential utility, I would really like to see the paper describe a bit more clearly what was actually done here, and how it compares with other methods of data gridding.**

**In particular, I think the authors should further explain and reference the neural network model used to generate the gridded product. There's only one paper in the references, from 1996, that seems to relate at all to the methods they applied. More detail should be given here so that the results could be reproduced, or extended as additional nitrate isotope data become available.**

**Next, the discussion and conclusions about the marine nitrogen cycle were largely confirmatory of earlier studies, but also almost beside the point of this particular manuscript. I would have found it more interesting, in the context of what was done here, to see how this kind of approach to data binning compares to alternative methods. Are there significant difference between this neural network approach, and a World Ocean Atlas approach of data interpolation? What are the implications of some of the choices made in building the model?**

**Specific comments are given below.**

**Lines 106-111: How does this neural network actually work? Does it use learning based on surrounding data to inform the values of unknown points? Where are the equations that go into the model? What is/are the function(s) that produces d15N values from the gridded T, S, NO3-, O2, and PO43- data?**
Our neural network has no explicit spatial component. We do not use latitude, longitude, or sampling depth as inputs to the model. Instead our model is purely a nonlinear function of physical and biological ocean parameters such as T, S NO3, etc. that all have implicit spatial characteristics. The model learns the relationship between d15N and these parameters for the locations where there are d15N

observations and, since we are using fields from the World Ocean Atlas (WOA) that have data everywhere, the model can estimate d15N for the locations where there are no observations using the nonlinear relationship it has learned. The function that models the relationship between d15N and training inputs is
d15N = a(a(I*W1+B1)*W2+B2)

where a is our activation function, which in this case is the hyperbolic tangent, I (size 7,170 binned observations by 6 input parameters) is our array of inputs [T S NO3 O2 ...], and W1 (size 6 by 25), W2 (size 25 by 1), B1 (size 25 by 1), and B2 (size 1 by 1) are our adjustable free parameters.

As a simple example, let us assume our only inputs (I) are T and S and they connect to a single node in the hidden layer. In this case, there are three total weights. One weight connects T to the hidden layer, one connects S, and another weight connects the hidden layer to the predicted d15N value. Let us also assume our activation function (a) is linear so we do not need to normalize our input data, and our bias weights (B1, B2) are zero. This simplifies the above equation to

d15N = (I*W1)*W2 = $(T*w_{11}+S*w_{12})*w_{21}$

For a single temperature and salinity pair (25 $^o$C, 33 PSU) and initial weights $w_{11}$ = 0.5 $^o$C$^{-1}$, $w_{12}$ = 0.5 PSU$^{-1}$, and $w_{21}$ = 0.2 permil

d15N = (25 *0.5 + 33*0.5)*0.2 = 5.8 permil. This is a predicted value. If our target value were 6 permil only small adjustments to the value of the weights would be necessary to match that observation. This works for a single observation. In reality, we have thousands of observations we would like to optimally match our predictions to, while at the same time not overfitting.

**Lines 116-119: Please clarify the description of depth binning.**
An observation is binned to the depth layer closest to its sampling depth. Observations with sampling depths at the midpoint between layers in the model grid are binned to the shallower layer. We have updated the manuscript accordingly.

**Lines 122-123: Why were whole ship tracks used for validation, rather than a more random selection?**
Our rationale for using whole ship tracks will be more clearly detailed in the revised manuscript and will be similar to the following response.

Note that this comment refers to our external validation, which is in addition to an internal validation that uses randomly selected data.

Imagine that we have a dataset that is made up of many cruises and we use a randomly selected 20% of this dataset for internal validation and another randomly selected 10% of this data to perform an external validation. Despite being randomly

selected, the external validating dataset will be from the same cruises as the wider data. In other words, despite being randomly selected, the validating dataset will be highly correlated geographically.

Instead, we have selected several cruises where none of the data was used to teach the model. These cruises are in areas where the model has not "learned" anything about nitrate and these data therefore provide a more difficult test of the model.

**Line 131: How was the daily chlorophyll used in an otherwise annual gridded product?**
We have updated the manuscript to clarify that daily chlorophyll data from the specified time period is not only binned to the model grid but also averaged to produce an annual field.

**Section 2.2 needs more references, especially 2.2.3 (lines 137-151). There is a lot of terminology here that is not defined or referenced, such as hidden layer, node, activation function, which should be defined and explained further. Also, it is not clear what you are applying weights to in the model.**
We have updated the text to provide a brief description of the neural network workflow, including defining some of the terms used and including a few additional citations. Weights form a linear system using inputs from the prior layer to produce values for the nodes in the next layer, as defined in a previous response. Using an activation function transforms this linear system to a nonlinear system. The hidden layer acts as intermediary between the input features and the target variable. Each of its nodes act as targets for the input layer and inputs for the final target layer. This increases the amount of learning the model can achieve by adding additional free parameters in the form of connections between nodes in one layer and nodes in the next.

**Line 158: It says that 10% of the observations were withheld to validate the networks. How were these chosen? More generally, how were the data for training, text, and validation chosen?**
We specify that 10% of the data is withheld randomly, but we updated the manuscript to clarify that EACH individual network has a random 10 percent withheld. This means each individual network sees a somewhat different training and test set. Some of the training data for one might be test data for another, and vice versa. Our final external validation set contains data that no individual network had available to it for training and is used to test the performance of the ensemble mean.

**Line 165: What are the implications of using whole cruise tracks for external validation rather than randomly chosen stations or grid cells?**
We responded to this above and will update the manuscript accordingly.

**Lines 179-180: Could this be shown (that the ensemble performs better than any single member of the ensemble) using your results, or is this a general**

**feature? Does it apply here?**
This is a general feature noted by Brieman (1996) that applies to certain machine learning methods such as EANNs. As our method uses EANNs, it applies here as well and the $R^2$ values of the internal validation sets versus the ensemble mean is greater than the $R^2$ value of each individual ensemble member because the ensemble mean incorporates members that saw different data during training. This does not necessarily apply to the external validation set, as that is comprised of data that no member has seen. However, the ensemble mean performs better than 19 out of 25 of the ensemble members on the external validation set in terms of a greater $R^2$ value and lower RMSE. Recall also that, since we curated ensemble members by first using the internal validation sets, these members are all higher performers, so the odds of roughly 1 in 5 of picking an ensemble member that does better on this particular external validation set is an overestimate of the actual odds if members were not curated. This is something that will be clarified in the updated manuscript.

**Discussion section:**
**How does the discussion stem from their results from the neural network model? Most of the discussion seems to focus on general features discussed in the original papers about the data used to generate the product. It would be more satisfying for this reviewer to read about how some of the choices they made in producing the model impacted the results.**
In order to reply to previous comments, the revised manuscript will necessarily have much more information on the inner workings of the model and how these choices impact the results. Hopefully these will address the immediate concerns of the Reviewer.

However, speaking as an observationalist (this is Rafter writing), I believe the most logical discussion of these modeling results requires an examination of how they fit with the published literature. As such, the Discussion section uses the model results to provide insight to marine nitrate δ15N that was previously hampered by poor geographic coverage.

**For example, 1) Is there only one way to produce the neural network model?**

1. A neural network model is a very general method, so there are many different ways to set up the architecture of the network, including number of hidden layers, size of hidden layers, how nodes in the hidden layer are activated, the type and number of input features we choose to include or not include, and the training algorithm among others. Aspects of these are covered by Rumelhart et al. (1986), Hornik et al. (1989), Weigand et al. (1990), and Thimm and Fiesler (1997).

**2) How were choices made? What tradeoffs were tolerated? What are the implications?**

2. The rationale for some of these choices were explicitly stated in section 2.2.3 of the manuscript, such as using only one hidden layer with 25 nodes in

order to keep the number of weights (free parameters) low relative to the number of training data, or our use of a hyperbolic tangent activation function.

Other choices were not explicitly stated and will be in the revised manuscript. For instance, the specific choice of our input features was dictated by our desire to achieve the best possible $R^2$ value on our internal validation sets. Additional inputs besides those we included, such as latitude, longitude, silicate, euphotic depth, or sampling depth either did not improve the $R^2$ value or degraded it, indicating that they are not essential parameters for characterizing this system.

Every choice was made for model simplicity, accuracy or a combination of the two. The inclusion of larger networks in terms of more input parameters resulted in models that did not generalize as well to new data, as indicated by their degraded performance on test data. Larger networks in terms of hidden layers and nodes increase each individual network's ability to learn on training data by virtue of there being more free parameters, but there is a general rule of how large a network should be relative to the amount of training data, as discussed by Weigand et al. (1990), and we tried to stay well within it.

One potential tradeoff is that other combinations of input features might better apply to certain regions than others. We opted to use the set of input features that yielded the best results globally, but on a regional scale other combinations of inputs may be better.

Having created a globally optimized, annual d15N climatology, there are several implications to consider. While, our external validation set demonstrates our model generalizes well to certain regions, it is clear that our model does not perform equally well everywhere. We opted for overall accuracy in our model, so for regions with relatively poor fit it is unclear whether this is due to our chosen combination of input features not working as well for a specific region or whether it is due to training data that is not representative of the mean state of d15N in that region.

**3) How does this approach compare with other methods for gridding?**

3. Standard interpolation techniques such as objective mapping would not work here, especially at 1-degree resolution and 33 vertical depth levels, due to the sparseness of the d15N data. Ocean parameters from the WOA, for instance, have much greater sampling density in order to create the interpolated fields. The EANN approach is more appropriate for sparse data, as it forms a relationship with more highly sampled ocean parameters to estimate d15N. There are many possible methods to model the relationship between these parameters and d15N, but simpler methods lack the complexity to adequately match the training data, let alone extrapolate well

to new data. As an example, we built a single global Multiple Linear Regression (MLR) model using all the same predictors used in the Ensemble Array of Neural Networks (EANN). We found that the MLR performs much worse than the EANN at predicting nitrate δ15N. The coefficient of determination for each method and each ocean basin's upper 1000 m is shown in the table below.

| | Atlantic | Pacific | Indian | Southern Ocean |
|---|---|---|---|---|
| MLR R2 | 0.04 | 0.49 | 0.51 | 0.34 |
| EANN R2 | 0.53 | 0.78 | 0.76 | 0.68 |

**4) Are there particular nodes that performed well in some locations vs. others?**

4. Because we randomly sampled from available observations to create the training data for each network, this sampling is pretty evenly distributed spatially. The same applies to test data. Since each network had to pass the same criteria on the test set in order to be admitted into the ensemble the individual networks do not greatly differ in their performance in regions where there is data, especially given that we optimized our combination of input parameters for a global analysis and did not consider different combinations that might lead to better regional performance.

   There are certain fairly large areas of the ocean where no observational data was available for this analysis. In these areas the individual ensemble members generate a larger range of estimates, as there is higher uncertainty about what the "truth" is. In these cases, the ensemble mean can be seen as splitting the difference or taking the most likely scenario of the estimates of d15N in these regions. That is the benefit of using the ensemble, as it provides the best general fit for the global ocean. The uncertainties of the EANN predictions are illustrated in Figure 5.

**Lines 415-423: It's not clear how the authors 'easily dismiss' an explanation about lateral advection of elevated nitrate d15N from ODZ regions. I think this section should be clarified. The way they set it up (seeing an increase in the Pacific but not Atlantic) does not seem to further the argument they are trying to make since the largest ODZ regions are in the Pacific, not the Atlantic.**

This discussion (which will be revised in the new manuscript) refers to deep Pacific nitrate $\delta^{15}$N, which increases from the Southern to Northern hemisphere. Similarly, deep Pacific waters originate at the Southern Ocean surface and move from the southern to northern hemisphere. An important addition to this discussion (suggested by Reviewer 1) is that while deep Pacific nitrate $\delta^{15}$N increases from

south-to-north, dissolved oxygen concentrations DECREASE and nitrate concentrations INCREASE. Grouping these observations together we have: (1) abyssal Pacific circulation moves from south-to-north, (2) oxygen decreases, (3) nitrate concentration increases, and (4) nitrate $\delta^{15}$N increases. Taken together, these known changes in deep Pacific waters are a persuasive argument that the change in deep Pacific nitrate $\delta^{15}$N originates from the remineralization of sinking organic matter (i.e., ammonification and nitrification of organic matter N).

The confusing part of this discussion (pointed out by the reviewer) is that this south-to-north elevation of deep Pacific nitrate $\delta^{15}$N cannot be explained by the lateral advection (i.e., along isopycnal) transport of high nitrate $\delta^{15}$N from the upper Pacific ODZ regions. This is because this explanation predicts that the highest nitrate $\delta^{15}$N would be found where shallow Pacific waters are first converted into deep Pacific waters in the deep South Pacific. Because this is the opposite of what we observe, this cannot explain the data.

**Figure 2. How many different selections of training, test, and validation sets did the authors test in the neural network model? What was the rationale behind the choice of the whole cruise tracks that were used for validation?**
This was answered above and new text will be available in the revised manuscript.

**Figure 3. Panel C was helpful. Panels A and B were also useful, but the choice of the non-linear color scale bar, where most of the data points were off scale, was unusual. In panel A, also please clarify whether this includes all of the data, or just those from the training set? Or validation set?**

We have adjusted the color bar in the revised manuscript (and see below). This figure includes all of the data where there are model results.

[Figure]

**Figure 4. The statistics for the different zonally averaged sections were useful, but I question the utility of the zonally averaged Pacific, given some of the large zonal gradients in d15N from the ODZs in the eastern tropical Pacific.**

We agree that they obscure the strong zonal gradients that occur in the lower latitude upper Pacific. But we also find them to be useful sources of discussion (for example the trends in deep Pacific nitrate δ15N). We will highlight the limitations of zonally-averaged view in the revised manuscript.

**Figure 5. The contours were extremely difficult to read, and the panels on the right hand side (E-H) were not particularly helpful. I also wondered how much of the patchiness, especially in panel A, is driven by the distribution of available d15N data?**

The revised Figure 5 can be seen below. We have discretized the color bar to more clearly indicate the contour value and use a color bar instead of black and white contours to show the standard deviation (right).

[Figure]

**Figure 6. Again, the contours are difficult to see. Can you indicate negative numbers with a different color, or allow the color bar to include negative numbers?**

The revised Figure 6 can be seen below. Once again we have discretized the color bar to more clearly illustrate the variability. We identify negative values by the dashed contour lines.